# Occurrence and Quantification of Porcine Hemotrophic Mycoplasmas in Blood-Sucking *Stomoxys calcitrans*

**DOI:** 10.3390/microorganisms13071607

**Published:** 2025-07-08

**Authors:** Mareike Arendt, Katharina Hoelzle, Julia Stadler, Mathias Ritzmann, Julia Ade, Ludwig E. Hoelzle, Lukas Schwarz

**Affiliations:** 1Institute of Animal Science, Department of Livestock Infectiology and Environmental Hygiene, University of Hohenheim, 70593 Stuttgart, Germany; ma.westenhoefer@gmail.com (M.A.); katharina.hoelzle@uni-hohenheim.de (K.H.); julia.ade@uni-hohenheim.de (J.A.); 2Clinic for Swine, Centre for Clinical Veterinary Medicine, Ludwig-Maximilians-Universität München, 85764 Oberschleissheim, Germany; j.stadler@med.vetmed.uni-muenchen.de (J.S.); ritzmann@med.vetmed.uni-muenchen.de (M.R.); 3Clinical Centre for Population Medicine in Fish, Pig and Poultry, Clinical Department for Farm Animals and Food System Science, University of Veterinary Medicine Vienna, 1210 Vienna, Austria; lukas.schwarz@vetmeduni.ac.at

**Keywords:** hemotrophic mycoplasmas, *Stomoxys calcitrans*, transmission

## Abstract

Hemotrophic mycoplasmas (HMs) are cell wall-less, small and uncultivable pathogens, which can cause infections in pigs with no to severe clinical signs and can contribute to significant economic losses in the pig industry. In addition to the known mechanical transmission routes of HMs (e.g., via blood-contaminated instruments or lesions from ranking fights), transmission to pigs by arthropod vectors such as *Stomoxys calcitrans* is being discussed. To date, there is scant available data concerning the transmission of HMs by stable flies. The objective of this study is to gain more data concerning the occurrence of HMs in *Stomoxys calcitrans*. Therefore, quantitative real-time PCR was conducted on different stable fly samples (surface washings and whole flies). We found *Mycoplasma* (*M.*) *suis* in 5.2% of crushed flies and 4.2% of fly wash solutions, and *M. parvum* was detected in 5.2% of flies and 9.4% of fly wash solutions. ‘*Candidatus* (*Ca*.) M. haemosuis’ was not detected in any sample. The mean bacterial loads were 2.0 × 10^2^ *M. suis*/fly, 9.3 × 10^2^ *M. suis*/fly wash solution and, for *M. parvum*, 2.4 × 10^3^ *M. parvum*/fly and 2.1 × 10^3^ *M. parvum*/fly wash solution. This molecular occurrence of porcine HMs in blood-sucking flies and reasonable bacterial loads in the two- to three-digit range demonstrate that these flies serve as mechanical vectors in stables and are, therefore, of epidemiological importance.

## 1. Introduction

Hemotrophic mycoplasmas (HMs), which appear worldwide, cause significant economic losses to the pig industry [1,2,3]. Most HMs are host specific and can occur in a wide range of animals [4,5,6,7,8,9]. In pigs, three HM species, *M. suis*, *M. parvum* and the recently described ‘*Ca*. M. haemosuis’, are known so far. *M. suis* causes infectious anemia in pigs (IAP), which can manifest itself in various degrees of severity, depending on host susceptibility and virulence [10]. Pigs with acute infections show high fever, hemolytic anemia and hypoglycemia, whereas chronic infections vary from subclinical courses to anemia in neonates, growth retardation in feeder pigs and poor reproductive performance and dysgalactia in sows [3,10,11]. Contrary to infections with *M. suis*, *M. parvum* seems to be non-pathogenic, and infections are characterized by the absence of clinical signs or anemia [12]. ‘*Ca*. M. haemosuis’ has been detected in pigs with and without clinical signs in China, Korea, Germany and Thailand [13,14,15,16,17,18]. Clinical signs described include high fever, reduced feed intake, skin alterations, increased mortality and a decreased daily weight gain [17].

It has been shown that pigs can be infected with HMs through subcutaneous, intravenous, intramuscular, intraperitoneal and oral inoculation under experimental conditions [10,19,20]. Blood-independent infection routes, such as via excretions, seem to play a minor role under field conditions [21]. However, mechanical blood transmission through contaminated instruments, in utero transmission to newborns and lesions from ranking fights play an important role in infections [20,22,23]. In addition, arthropod vectors (e.g., stable flies, mosquitos, pig lice) are suspected to be involved in the transmission of porcine HMs. Though there is less data about the transmission of HMs by stable flies, in an infection experiment, healthy pigs were infected by stable flies (*Stomoxys calcitrans*) that had previously been blood-fed on *M. suis* infected pigs [23,24]. The obligate blood-sucking *Stomoxys calcitrans*, which can be found all over the world and is often present in pig farms, is also known to contribute to the mechanical transmission of other pathogens, such as Capripox virus, African swine fever virus, lumpy skin disease virus and *Bacillus anthracis* [25,26,27,28,29]. Due to defensive movements of the host after biting or due to other flies, the stable fly frequently changes hosts to continue its blood meal. Consequently, pathogens can be transferred, for instance, through their mouthparts or by regurgitation [30,31,32].

The aim of this study was to investigate the occurrence and bacterial loads of uncultivable HMs in stable flies (*Stomoxys calcitrans*) and to gain more knowledge about the vector potential of *Stomoxys calcitrans* for porcine HMs.

## 2. Material and Methods

### 2.1. Samples and Sample Preparation

Stable flies (*Stomoxys calcitrans*) from 20 Austrian piglet producing farms (stables 1–20) were available from a previous study [28]. Samples were collected mainly from the gestation area of farms with and without disease problems and subsequently screened for different bacteria and viruses including hemotrophic mycoplasma. Three flies from each pig farm were analyzed in this study.

Additional samples that originated from five Austrian pig farms with unknown HM status (stables 21–25), consisting of 10 blood samples and 6 flies per stable, were examined.

The Austrian farms were located in Lower Austria, Upper Austria, Styria and Vienna.

Furthermore, 6 stable flies from an HM-positive (*M. suis*, *M. parvum* and ‘*Ca.* M. haemosuis’) farm in Northern Germany, North Rhine-Westphalia, (stable 26) were tested for HMs.

Each stable fly sample was first washed in 400 µL Dulbecco’s Phosphate Buffered Saline (Biowest, Nuaillé, France). Then, the flies were individually washed in 70% ethanol, air dried and crushed with micropistills in a sterile 1.5 mL tube. DNA was isolated from the wash solutions and crushed flies using the GenElute^TM^ Bacterial Genomic DNA Kit (Sigma Aldrich, Darmstadt, Germany) according to the user’s manual, except for the following modification for crushed stable flies: Step 1 (centrifugation) was omitted, and the homogenized flies were mixed directly with 180 µL Lysis Solution T.

DNA from EDTA blood samples was extracted according to the manufacturers protocol using the Monarch^®^ Genomic DNA Purification Kit (New England Biolabs, Frankfurt am Main, Germany) with the following preliminary steps to gain better HM DNA yield: a total of 200 µL EDTA blood was mixed with 200 µL lysis buffer (10 mM Tris-HCL pH 7.5, 5 mM MgCl_2_, 0.32 M saccharose, 1% (*v*/*v*) Triton X-100) and then centrifuged at 8000 rpm for 1 min. The supernatant was discarded. The pellet was resuspended in 400 µL lysis buffer and centrifuged as described above. This step was repeated once.

### 2.2. Quantitative Real-Time PCR

*M. suis*, *M. parvum* and ‘*Ca.* M. haemosuis’ were detected using the StepOne^TM^ System (Applied Biosystems, Darmstadt, Germany). For detection of *M. parvum*, quantitative real-time PCR was performed as described elsewhere using primers *MPa*F and *MPa*R [33]. The samples were screened on *M. suis* by using the SYBR green assay and primers *Msg*1-Fw and*Msg*1-Rv targeting the MSG1 gene of *M. suis* as described by Hoelzle et al. [34]. ‘*Ca.* M. haemosuis’ was detected by quantitative real-time PCR using primers *CMhsuis*F and *CMhsuis*R to amplify a 177 bp fragment of the *gap* gene [35]. In each PCR, 2 µL DNA template was added to 18 µL PCR mixture containing 10 µL Fast SYBR^TM^ Green Master Mix (Thermo Fisher Scientific, Darmstadt, Germany) and 8 µL primer mixture (containing 0.5 µM primer each for *M. suis* and ‘*Ca.* M. haemosuis’ and 0.4 µM primer each for *M. parvum*). Quantification was performed using plasmid DNA of *M. suis*, ‘*Ca.* M. haemosuis’ and *M. parvum* with standard dilutions 10^5^, 10^4^ and 10^3^ genome equivalent (GE) per reaction in each qPCR run. By applying the Standard Curve Method of the StepOne^TM^ software version 2.2 (Applied Biosystems), the obtained Ct values were then extrapolated into HM GE/reaction [33,34,35].

Further data analysis was performed using StepOne^TM^ software version 2.2 (Applied Biosystems) and Microsoft^®^ Excel, 2016.

### 2.3. Next Generation Sequencing

Two crushed fly samples, one that tested positive for *M. suis* and one that tested positive for *M. parvum*, were sent to Microsynth AG (Balgach, Switzerland) for next generation sequencing-based 16S rRNA analysis and subsequent data analysis including figure creation. Therefore, washing of flies and DNA isolation was performed as described above. Library preparation and sequencing, using the amplicon metagenomics approach, were then performed by Microsynth AG. By using the Illumina MiSeq system (Illumina, San Diego, CA, USA), they sequenced the hypervariable regions on the 16SrRNA gene, using universal primers 341F and 805R [36].

## 3. Results

### 3.1. Detection of HM Species by Quantitative Real-Time PCR

*Mycoplasma suis* was found in 4 out of 96 fly wash solutions (4.2%) and in 5 out of 96 fly samples (5.2%) originating from a total of six Austrian pig farms. Bacterial loads ranged from 7.7 × 10^2^ to 1.1 × 10^3^ *M. suis* in total positive wash solution samples (mean value 9.3 × 10^2^ *M. suis*/washing solution sample) and 2.9 × 10^1^ to 3.8 × 10^2^ bacteria in positive fly samples (mean value 2.0 × 10^2^ *M. suis*/fly sample). The results of *M. suis*-specific real-time PCRs are depicted in Table 1.

A total of 9 out of 96 wash solution samples (9.4%) tested positive for *M. parvum*, and 5 out of 96 fly samples (5.2%) were shown to be *M. parvum* positive. The *M. parvum* positive samples originated from a total of 12 stables. Bacterial loads ranged from 3.4 × 10^2^ to 5.9 × 10^3^ *M. parvum*/total washing solution sample (mean value 2.1 × 10^3^ *M. parvum*/washing solution sample) and, in fly samples, from 1.2 × 10^3^ to 3.9 × 10^3^ *M. parvum*/fly sample (mean value 2.4 × 10^3^ *M. parvum*/fly sample). The results of *M. parvum*-specific real-time PCRs are summarized in Table 2.

In 4 out of 26 stables (15.4%), both *M. parvum* and *M. suis* were detected in *Stomoxys* samples. In a total of 6 out of 26 stables (23.1%), *M. suis* positive flies were detected. *M. parvum* positive flies were found in 12/26 (46.2%). In HM positive stables, 28.9% (13/45) of the wash solution samples and 22.2% (10/45) of the fly samples tested positive.

‘*Candidatus* M. haemosuis’ was not detected in any of the *Stomoxys* samples.

### 3.2. Sequence Analyses

The results of the next generation sequencing-based 16S rRNA analysis (NGS) approach are summarized in Table 3. No HM specific sequences were identified by 16S rDNA NGS; only the presence of other bacteria, including *Morganella morganii* and *Arthrobacter russicus*, was detected.

## 4. Discussion

The epidemiology of porcine HM infections is rather unknown, and transmission by vectors is a recurring topic of discussion. The hematophagous stable fly *Stomoxys calcitrans* is often found in pig stables. Several studies have shown that *Stomoxys calcitrans* can transmit pathogens by mechanical transmission, carrying the pathogens on the body surface or via blood-feeding acts [31,37]. In the present study, we investigated *Stomoxys calcitrans* for the presence of the three porcine HM species *M. suis*, *M. parvum* and ‘*Ca.* M. haemosuis’ using species-specific real-time PCR assays. Because HMs are uncultivable, cultural methods cannot be considered for detection. The two HM species *M. suis* and *M. parvum* were present in the investigated flies, indicating that *Stomoxys calcitrans* could serve as a vector for these pathogens. Schwarz and co-workers examined pools of 10 flies out of some of the same stables (stables 1–20) for the presence of different pathogens including HMs. Using an HM universal PCR and subsequent sequence analyses, they identified 35% of the pig farms (7/20 farms) as positive for HMs [28]. The detected HM species also included *M. suis* and *M. parvum*. In this study, one farm with HM-positive stable flies had a history of IAP. However, at the time of sampling, the animals did not show any related clinical signs. In our study, HMs were found in 13 out of 26 stables (50%). In 4 of the 26 stables investigated, both *M. parvum* and *M. suis* were present in the flies. Differences in the detection rates between the present study and the study of Schwarz et al. [28] are most likely due to the investigation of single flies in our study instead of fly pools and the use of highly sensitive and species-specific real-time PCR assays. In our study, six fly samples from an HM-positive (*M. suis*, *M. parvum* and ’*Ca.* M. haemosuis’) farm in Northern Germany (stable 26) tested negative for HMs, indicating that large sample sizes of flies need to be analyzed to obtain information about the HM status of stables.

We analyzed surface wash solutions and whole flies in order to gain some insights into potential transmission modes. *M. suis* was present in 4.2% (4/96) of surface wash solutions and in 5.2% (5/96) of fly samples. *M. parvum* was present in 9.4% (9/96) of wash solutions and 5.2% (5/96) of whole *Stomoxys calcitrans* samples.

Positive wash solutions show that HMs are located on the surface of flies. There are essentially two possible ways on which these bacteria can get onto the surface of flies. Bacteria could either be transferred to the fly surface directly from infected pigs, or the fly surface could be contaminated via regurgitation of saliva or via fly feces. Furthermore, *M. suis* and *M. parvum* were detected in whole flies after thorough washing, indicating that the HMs are present in parts of the flies’ bodies. These results show that there are different possible modes of transmission. During the feeding act, *Stomoxys* can restart their blood meal on another host and hereby transmit pathogens from the surface of the fly into lesions of livestock [32]. Also, when injecting saliva prior to blood-sucking, they can inoculate with infected blood remaining on their mouthparts or regurgitated pathogens from their crop [31,38].

Interestingly, *M. parvum* was found more often in wash solutions than in the washed flies’ bodies, whereas for *M. suis*, the amounts of positive whole fly samples and wash solutions were similar. However, as the number of samples is very small, no final statement can be made as to whether there is a difference between the two species *M. suis* and *M. parvum* or whether the difference is a random result.

‘*Candidatus* M. haemosuis’ was not detected in any sample. Schwarz and co-workers were also not able to detect ‘*Ca.* M. haemosuis’ in the stable flies examined. However, the negative results regarding ‘*Ca.* M. haemosuis’ do not exclude the potential vector role of *Stomoxys calcitrans*, as ‘*Ca*. M. haemosuis’ is an emerging pathogen that has only been detected in domestic pigs from Asia (China, Republic of Korea and Thailand) and Germany so far [15,16,17,18,35]. In Austria, ‘*Ca.* M. haemosuis’ has not been confirmed yet, indicating a minor significance of this species in Austria so far. The overall lower prevalence of ‘*Ca*. M. haemosuis’ could be an explanation for the lack of detection of ‘*Ca*. M. haemosuis’ in the *Stomoxys* samples analyzed in the present study.

Arthropod contact as a potential transmission route for porcine HM is also supported by a previous study from Song and co-workers. The study proved that the prevalence of *M. suis* in summer and autumn was significantly higher than in other seasons. Thus, the air humidity in these seasons is more favorable for insects, and the study concluded that arthropod vectors might play a role in the transmission of *M. suis* [39]. In addition to the above-mentioned study by Schwarz and co-workers [28], only one previous study investigated the potential of *Stomoxys* in the experimental transmission of porcine HMs. Prullage and co-workers showed, in their infection study, that *Stomoxys calcitrans* fed on *M. suis*-infected pigs were able to infect splenectomized pigs when transferred immediately [24]. With longer intervals between feeding and contact with splenectomized pigs, the pigs could no longer be infected. Due to their lack of ability to maintain the pathogen, the study also concluded that *Stomoxys calcitrans* flies serve as mechanical vectors. In contrast, biological vectors can transmit pathogens after significant delay, as they are able to maintain the pathogen [24]. We could prove that HMs can be found in wash solutions of fly samples and crushed fly samples. Therefore, our results also suggest that *Stomoxys calcitrans* might serve as a mechanical vector for *M. suis* and *M. parvum* and transmit the bacteria either via surface contact or during the blood meal.

All other studies regarding the HM vector potential of *Stomoxys calcitrans* in livestock investigated bovine, ovine and buffalo HM species. One previous study detected the bovine HM species *M. wenyonii* in 41.41% of stable flies collected from a buffalo farm in Thailand [18]. Hornok and co-workers detected *M. wenyonii* and ‘*Ca.* M. haemobos’ DNA in *Stomoxys calcitrans* flies collected from a stable with HM-infected cattle [40]. One experimental study on ovine HMs was conducted by Overas in 1969, where *Stomoxys* spp. were used to successfully transfer *M. ovis* from infected to uninfected sheep [41]. However, in this study, maintenance and propagation of HMs in the vector were not included. Most studies investigating the potential of *Stomoxys calcitrans* as an HM vector for livestock are based on molecular detection methods due to HMs being uncultivable [42]. Although there was no evidence of experimental HM transmission in these studies, the detection of the pathogens in potential vectors nevertheless provides evidence of vector potential [43]. To underline the potential of *Stomoxyx calcitrans* as mechanical vectors, additional studies combining experimental studies and molecular methods such as quantification of porcine HMs in flies with different starvation intervals after blood meals should be considered in the future. The quantification of HM DNA in the flies examined revealed mean bacterial loads of 3.9 × 10^2^ *M. suis* and 2.1 × 10^3^ *M. parvum* for the total washing solutions and of 2.0 × 10^2^ *M. suis* and 2.4 × 10^3^ *M. parvum* in the crushed fly samples, respectively. To the best of our knowledge, this is the first molecular quantification study of porcine HMs in *Stomoxys* species. Similar quantities were found for *M. wenyonii* in *Stomoxys* samples from a Hungarian cattle herd, with a mean copy number of 1.0 × 10^2^. The same study, however, revealed notably lower copy numbers for ‘*Ca.* M. haemobos’ (mean copy number of 1.0) in stable flies [40]. Based on their results, the authors suggested that the bacterial loads in hematophagous arthropods may be influenced by (1) the level of the bacteremia in livestock and, thus, the HM availability for potential blood-sucking vectors or (2) the differences in the transmission efficiency between HM species [40]. In the present study, HM blood levels of pigs at the time of fly sampling in the respective stables are unfortunately not available. Another factor influencing the bacterial load in the flies might be the decrease in bacteria with temporal distance from the blood meal. Prullage found that the transmission capability of *Stomoxys calcitrans* for *M. suis* is no longer present 24 h after the last blood meal [24]. Studies with other pathogens such as West Nile virus in stable flies also showed a rapid decrease in the number of flies containing the virus and the amount of virus within flies with distance to the blood meal [44]. Therefore, the bacterial loads detected in *Stomoxys calcitrans*, tested immediately after blood meals, may be different. In the present study, information on the time of the last blood meal is not available. However, several previous studies demonstrated that HMs can also be transmitted by hematophagous arthropods, even at low bacterial loads. In the case of *M. ovis*, it was shown that one infected red blood cell is sufficient for mechanical transmission. Consequently, *M. ovis* infection through natural vector-borne transmission can be initiated at a low level of parasitemia in sheep along with a high-level activity of blood-sucking insects [45].

In a second approach, we used high-throughput 16S rRNA amplicon sequencing as an alternative method for the investigation of *Stomoxys calcitrans* as a vector for HMs. This approach has the advantage over PCR-based methods in that not only known species can be detected but also those previously unknown. However, it was not possible to detect HM sequences by next generation amplicon sequencing in *M. suis* or *M. parvum* qPCR positive samples. Obviously, the HM loads in the *Stomoxys* samples investigated were too low. Rather than HM sequences, NGS results revealed mainly *Morganella morganii* ssp., *Arthrobacter russicus* and *Phenylobacterium koreense* sequences. The presence of various other bacteria in stable flies in combination with universal primers seems to result in superimposing the presence of the small amounts of HMs. This emphasizes that, for detecting small amounts of HMs, very specific methods, such as quantitative real-time PCR, are compulsory. The detection of the above-mentioned bacterial species was not surprising, since Schwarz et al. also detected *Morganella morganii* and *Arthrobacter* sp. by culture and matrix-assisted laser desorption ionization–time of flight mass spectrometry (MALDI-TOF MS) in fly samples out of the same Austrian pig stables (stables 1–20) [28].

## 5. Conclusions

The detection of hemoplasma DNA (*M. parvum* and *M. suis*) in stable flies indicates their potential to contribute to the spreading of HMs. Vector transmission through widespread *Stomoxys calcitrans* could possibly have an epidemiological impact, since it could allow for intra- and inter-herd HM spread. Since *M. suis* and *M. parvum* are highly prevalent in wild boars, HM vector transfer from the wildlife reservoir to pig farms should also be considered regarding biosafety measures. Moreover, *Stomoxys calcitrans* might be used to monitor the hemoplasma status in the stable, but the absence of HMs in stable flies does not allow for conclusions regarding the absence of HMs in pigs. Detected bacterial loads of HMs in stable flies are rather low. Therefore, precise and highly sensitive detection methods are required to prove the presence of HMs in stable flies. Further studies are called for in order to gain more knowledge on the role of *Stomoxys calcitrans* as a vector for HMs, such as experiments with larger numbers of flies and blood samples of HM positive and negative stables or quantification of HMs in flies with different starvation intervals after blood meals, to obtain information about persistence duration within the vector.

## Figures and Tables

**Table 1 microorganisms-13-01607-t001:** Results of *M. suis*-specific real-time PCRs.

Sample Source	Wash Solution		Fly	
	Positive/All Tested	Copy Number in Total Positive Wash Solution	Positive/All Tested	Copy Number in Positive Fly
stable 4	0/3	n.d.	1/3	2.9 × 10^1^
stable 6	1/3	8.4 × 10^2^	1/3	3.8 × 10^2^
stable 7	1/3	7.7 × 10^2^	1/3	8.0 × 10^1^
stable 8	0/3	n.d.	1/3	2.6 × 10^2^
stable 13	1/3	1.1 × 10^3^	1/3	2.5 × 10^2^
stable 16	1/3	1.1 × 10^3^	0/3	n.d.

**Table 2 microorganisms-13-01607-t002:** Results of *M. parvum*-specific real-time PCRs.

Sample Source	Wash Solution		Fly	
	Positive/All Tested	Copy Number in Total Positive Wash Solution	Positive/All Tested	Copy Number in Positive Fly
stable 6	1/3	1.8 × 10^3^	0/3	n.d.

stable 7	1/3	1.7 × 10^3^	0/3	n.d.
stable 8	2/3	7.6 × 10^2^	0/3	n.d.
		1.8 × 10^3^		
stable 9	1/3	9.0 × 10^2^	0/3	n.d.
stable 10	1/3	1.6 × 10^3^	0/3	n.d.
stable 14	0/3	n.d.	1/3	3.3 × 10^3^
stable 15	1/3	3.4 × 10^2^	0/3	n.d.
stable 16	0/3	n.d.	1/3	1.9 × 10^3^
stable 17	0/3	n.d.	1/3	1.9 × 10^3^
stable 18	1/3	4.6 × 10^3^	0/3	n.d.
stable 19	1/3	5.9 × 10^3^	1/3	3.9 × 10^3^
stable 22	0/6	n.d.	1/6	1.2 × 10^3^

**Table 3 microorganisms-13-01607-t003:** Results of next generation sequencing.

Operational Taxonomic Unit (OUT)	Bacteria	Presence (%)	
ID		Sample 1	Sample 2
1	*Morganella morganii* subsp. *morganii*	30.1	0.5043
2	*Morganella morganii* subsp. *morganii*	13.94	0.1441
3	*Morganella morganii* subsp. *sibonii*	13.93	0.2161
4	*Morganella morganii* subsp. *morganii*	13.82	0.07205
5	*Morganella morganii* subsp. *sibonii*	13.33	0.07205
6	*Morganella morganii* subsp. *sibonii*	13.26	0.2161
7	*Arthrobacter russicus*	1.582	96.61
8	*Phenylobacterium koreense*	0.02561	2.161

## Data Availability

The original contributions presented in this study are included in the article. Further inquiries can be directed to the corresponding author.

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
