# Peer review of "Occurrence and Quantification of Porcine Hemotrophic Mycoplasmas in Blood-Sucking Stomoxys calcitrans"

_microorganisms, 2025, doi:10.3390/microorganisms13071607_

Round 1
Reviewer 1 Report
Comments and Suggestions for Authors
The manuscript addresses a relevant topic with appropriate methods and presents some interesting results. The major flaw is a general lack of rigour and focus, especially in the discussion. One reason for this could be the weak writing style in English (see below).
The 16S rDNA NGS approach was not successful, so the detailed presentation of the data seems a bit random and distracting. In my opinion, the negative results should be presented without showing the Figures 2 and 3.
Abstract
L15: Change „none“ to „no”
L16: Rewrite: “Apart from the known routes blood-borne transmission… ”
L17: Change by ranking fight” to “from ranking fights”
L17: Change “transfer on pigs” to “transfer to pigs”
L18: Change “ until now” to “to date, …”
L18: Change “is little” to “are few”
L18: Change “about” to “on”
L19: Change “about” to “on”
L20: Change “to” to “on”
L21: Change “various” to “different”
L22 ff.: Add “of” after each percentage number
L26: Change “evidenced” to “proved”
L27: Change “those” to “these”
Methods
L69: Delete “of the remaining” because this information does not make much sense here
L70: Change “of” to “from”
Delete Figure 1. The information content is sparse and can easily be expressed in half a sentence. Also, the colouring is misleading.
L82: Change ”Consequently” to “Subsequently”
L86: Change “left out” to “omitted”
L106-107: A brief explanation of how the quantification was performed would save the reader a tedious search of the secondary literature.
L111-113: Which samples were used: flies or washings? Any specific preparation of DNA? There is almost no information on the sequencing method used. NGS can be so many very different things…
Results
Sum of fly samples 3x20+6x5+6x1=96 (according to Material and Methods L66-74, but total sample number is given with 90 (L117, L125)???
L141-133, Table 3 and Fig 2 and 3: The figures are cryptic and do not seem to agree with the table.
Discussion
L177-190: This paragraph is not well written and the argumentation is not clear. The expectation is that HMs are transmitted by blood and the blood-sucking process. Therefore, it is unexpected that the flies carry the pathogens also on their surfaces. The authors speculate for the reasons why. They should also try to explain how the HMs can survive longer on the flies’ surface even though they depend on blood for nutrition and survival.
L191-205: This paragraph is too long and bulky to discuss the simple fact that M. haemosuis was probably not detected because it is generally less prevalent in pigs compared to the other two HMs.
L204: Change “the missing” to “the lack of”
L218-219: Change ”Due to their missing capability of maintaining the pathogen in the vector the study also concluded that Stomoxys calcitrans serve as mechanical vectors” to “Due to their lack of ability to maintain the pathogen, the study also concluded that Stomoxys calcitrans flies serve as mechanical vectors”
L222-233: Also this paragraph lacks a clear structure and message.
L234: Change “M. suis” to “HM DNA” or “HM genome equivalents (GE)”
L242: Add “in livestock” after “the level of the bacteremia”
L247: Change “with distance to blood feeding” to “with temporal distance from the blood meal” (correct also in L251)
L252: “and thus from the infectious dose” – This makes no sense in this sentence. Please, rewrite.
Conclusion
L276-277: Delete “to carry over diseases and” because this conclusion is not correct. The flies can potentially only transmit pathogens, but not disease.
L277-278: It is not correct to state that “Vector transmission through widespread Stomoxys calcitrans has obviously a considerable epidemiological impact.” Because this was not shown, but that it COULD HAVE (potentially have an impact.
L286: Could you please specify which “further studies” you mean?
Comments on the Quality of English LanguageIt is strongly recommended to revise the manuscript with a native speaker or with the help of appropriate language tools (DeepL or others). I have made some corrections, at least for the abstract. For the other parts of the manuscript, you will have to make further corrections yourself. In general, to improve readability, please put a comma after introductory or subordinate clauses, phrases or words.
Reviewer 2 Report
Comments and Suggestions for Authors
In the manuscript titled “Occurrence and quantification of porcine hemotrophic mycoplasmas in blood - sucking Stomoxys calcitrans” authored by Mareike Arendt et al., the researchers utilized qPCR as a means to detect the bacterial loads of porcine hemotrophic mycoplasmas (HMs) both internally and externally harbored by Stomoxys calcitrans, and aimed to evaluate the potential role of Stomoxys calcitrans as mechanical vectors for HMs within porcine stable environments.This manuscript is very meaningful, but there are some problems that need to be pointed out.
1.While this study employs qPCR to screen HMs in external washes and whole-body homogenates of Stomoxys calcitrans, critical experimental evidence remains lacking to conclusively demonstrate mechanical transmission capability. The experimental design should incorporate three essential components: (1) Current detection methodology focusing on superficial washes and whole-fly homogenates fails to distinguish between passive contamination and active transmission potential. (2) Microbial culture isolation from internal compartments (particularly the proboscis and digestive tract) to confirm retained pathogen viability post-feeding, and (3) A time-course analysis using controlled starvation intervals post-blood meal to quantify HMs persistence duration within the vector. These additions would provide crucial evidence for both mechanical transfer capacity and temporal transmission windows.
2.The manuscript inadequately addresses the critical epidemiological parameter of vector infection rates across different farm statuses. A comparative analysis of HMsdetection frequency in flies from HMs-positive versus negative farms is essential to establish any statistically significant association. Particularly problematic is the null result from Stable 26 (HMs-positive farm in Northern Germany), where all 6flies tested negative despite confirmed farm-level pathogen presence. This discrepancy directly challenges the proposed transmission hypothesis and necessitates either: (a) Expanded sample sizes to improve detection power, or (b) Consideration of alternative explanatory factors (e.g., vector density thresholds, host bacteremia levels, or environmental persistence parameters) that might modulate transmission efficiency.
Reviewer 3 Report
Comments and Suggestions for Authors
The authors aimed to envestigate the occurrence of three bacterial species transmitted mechanically or vectorially by stable flies of the species Stomoxys calcitrans (Muscidae).
To investigate the occurrence and bacterial loads of hemotrophic mycoplasmas (HMs) Mycoplasma suis, Mycoplasma parvum, and Candidatus Mycoplasma haemosuis transmitted by stable flies, the authors used real-time PCR assays.
The use of body wash water from flies as a means to detect potentially mechanically transmitted bacteria is, in my view, a very interesting approach.
The results are clearly presented, and in the discussion, the authors acknowledge certain limitations, such as the sample size, which should be increased to enable comparisons with other studies. As a distinguishing feature compared to other studies, the authors highthey the gap in research focused on the epidemilogy of swine hemotrophic mycoplasmas. They emphasize the individual investigations of flies regarding their potential as both biological and mechanical vectors, through the analysis of the flies body wash solutions.
In my opinion, it is not advisable to include references in the conclusion section.
Round 2
Reviewer 1 Report
Comments and Suggestions for Authors
The efforts of the authors to improve the manuscript is acknowledged. In my opinion it has improved, however, small corrections are still needed before publication.
L113: Delete "by" before "using"
L114: Write "genome equivalent (GE)
L158: Chang "was" to "were"
L120 and L156: Change "Next generation sequencing" to " Next generation sequencing-based 16S rRNA analysis"
L185: comma after "In our study"
L194-197: Move this paragraph right after L189.
L237: Comma after [24].
L263-265: Remove this information here. It is redundant with L239-241.
L274: Add "the" after "in".
L259: Add "HM" before "vector".
L260-261: Remove the added sentence because it is redundant and breaks the line of argumentation.
L323: Change "a" to "an".
L324: Comma after wild boar.
L331: Remove comma after "called for".
Comments on the Quality of English LanguageIt is still not perfect, but acceptable.
